# Feasibility of Comprehensive Genomic Profiling (CGP) in Real-Life Clinical Practice

**DOI:** 10.3390/diagnostics13040782

**Published:** 2023-02-19

**Authors:** Lorenzo Nibid, Giovanna Sabarese, Daniela Righi, Silvia Maria Rossi, Giorgia Merlini, Pierfilippo Crucitti, Bruno Vincenzi, Giuseppe Tonini, Giuseppe Perrone

**Affiliations:** 1Research Unit of Anatomical Pathology, Department of Medicine and Surgery, Università Campus Bio-Medico di Roma, Via Alvaro del Portillo, 21-00128 Roma, Italy; 2Anatomical Pathology Operative Research Unit, Fondazione Policlinico Universitario Campus Bio-Medico, Via Alvaro del Portillo, 200-00128 Roma, Italy; 3Research Unit of General Surgery, Department of Medicine and Surgery, Università Campus Bio-Medico di Roma, Via Alvaro del Portillo, 21-00128 Roma, Italy; 4Thoracic Surgery Operative Research Unit, Fondazione Policlinico Universitario Campus Bio-Medico, Via Alvaro del Portillo, 200-00128 Roma, Italy; 5Research Unit of Oncology, Department of Medicine and Surgery, Università Campus Bio-Medico di Roma, Via Alvaro del Portillo, 21-00128 Roma, Italy; 6Medical Oncology Operative Research Unit, Fondazione Policlinico Universitario Campus Bio-Medico, Via Alvaro del Portillo, 200-00128 Roma, Italy

**Keywords:** NGS, CGP, F1CDx, solid tumors, molecular biology, feasibility

## Abstract

In advanced or metastatic settings, Comprehensive Genomic Profiling (CGP) allows the evaluation of thousands of gene alterations with the goal of offering new opportunities for personalized treatment in solid tumors. This study evaluated the CGP *Success Rate* in a real-life cohort of 184 patients enrolled in a prospective clinical trial. CGP data were compared with the routine molecular testing strategy adopted in-house. Sample age, tumor area, and the percentage of tumor nuclei were recorded for CGP analysis. We found that 150/184 (81.5%) samples resulted in satisfying CGP reports. The CGP *Success Rate* was higher in samples from surgical specimens (96.7%) and in specimens that had been stored (sample age) for less than six months (89.4%). Among the inconclusive CGP reports, 7/34 (20.6%) were optimal samples, according to CGP sample requirements. Moreover, with the in-house molecular testing approach, we could obtain clinically relevant molecular data in 25/34 (73.5%) samples that had inconclusive CGP reports. In conclusion, despite the fact that CGP offers specific therapeutical options in selected patients, our data suggest that the standard molecular testing strategy should not be replaced in routine molecular profiling.

## 1. Introduction

Testing for genomic alterations is essential for selecting the most appropriate treatments and in clinical trials. Once genomic alterations are known, one can avoid applying futile therapies when resistant tumors are present or actionable targets are absent [1,2].

The ability to apply precision medicine in treating patients with cancer requires an approach that can identify targetable alterations, which can occur in many genes, in many types of alterations, and in complex genomic signatures. With the proliferation of highly effective targeted therapies, it may not always be possible to determine the best treatment by examining the patient or running a single analytic test; rather it may require a molecular diagnostic, which is both time- and tissue-efficient, that can also deliver accurate, reproducible performance. The advent of New-Generation Sequencing (NGS) revolutionized the clinical practices involved in diagnosing and treating cancer. The application of NGS to a wide panel of genes is known as Comprehensive Genomic Profiling (CGP) [3,4].

FoundationOne®CDx (F1CDx, Foundation Medicine, Cambridge, Massachusetts, USA) is a companion diagnostic test that was approved by the United States Food and Drug Administration. F1CDx aims to identify patients that may benefit from treatment with 28 specific drug therapies, in accordance with the approved therapeutic product labeling. F1CDx utilizes NGS-based CGP technology to examine 324 cancer genes in solid tumors. The analysis is performed in centralized laboratories and the results identify known and likely pathogenic short variants (SVs), copy number alterations, and select rearrangements. In addition, CGP employs complex biomarkers to investigate the tumor mutational burden (TMB), microsatellite instability (MSI), and genomic loss of heterozygosity, for example, in ovarian cancer [4]. Nevertheless, CGP analysis is not currently adopted worldwide as a standard of care for solid tumors. In fact, in current clinical practice, mutational status is evaluated through in-house tests that focus on single-target genes (e.g., quantitative PCR (q-PCR), real-time PCR (RT-PCR), fluorescence in situ hybridization (FISH), or immunohistochemistry (IHC)) and/or on panels of a few genes (e.g., RNA-NGS or DNA-NGS), depending on the resources of the local laboratory [5,6].

To establish the feasibility of CGP as a standard of care for solid tumors, here, we investigated the CGP approach in real-life samples from a prospective cohort of patients. Then, CGP data were compared with those obtained from in-house molecular tests.

## 2. Materials and Methods

CGP data were obtained from F1CDx reports of patients enrolled between November 2019 and October 2021 in the prospective clinical trial GO40782–RXDX−101–02. Medical oncologists selected patients to be tested by CGP on the basis of clinical evaluation (patients that had no other approved therapy available and patients that did not respond to standard therapies, according to their clinical, pathological, and molecular characteristics) and availability of tumor tissue.

Each tumor sample was formalin-fixed and embedded in a paraffin block. Samples were analyzed with an F1CDx assay. According to F1CDx-sample requirements, we classified tumor samples as “optimal”, “acceptable”, and “suboptimal”, based on the surface area (SA), and the percentage of tumor nuclei (TN%). SA and TN% were evaluated by two pathologists (GP, LN) that reviewed hematoxylin and eosin (H&E)-stained slides.

We classified a sample as “optimal” when both the SA and TN% were evaluated as optimal, according to Foundation Medicine criteria. We classified a sample as “acceptable” when both the SA and TN% were evaluated as acceptable, or when one parameter was optimal and the other one was acceptable. We classified a sample as “suboptimal” when at least one parameter (SA and/or TN%) was evaluated as suboptimal.

For each sample, the date of sample collection at the Pathology Department and the date of CGP analysis were extracted. Based on these dates, we define the “sample age” as:Sample age = CGP analysis date − sample collection date(1)

Sample ages were categorized into 4 classes: <6 months, 6–12 months, 12–24 months, and >24 months.

Moreover, CGP reports were classified as (a) *completely satisfying* when they contained complete genomic data, including data on MSI and TMB; (b) *partially satisfying* when they contained complete genomic data, but not data on MSI or TMB; and (c) *inconclusive* when they contained no genomic results. Based on these classes, we defined the “CGP *Success Rate*” as follows:(2)$$\mathrm{CGP}\;\mathrm{Success}\;\mathrm{Rate}=\frac{Completely\;satisfying\;reports+Partially\;satisfying\;reports}{Total\;number\;of\;patients}\times 100$$

We investigated correlations between the CGP *Success Rate* and sample age, SA, and TN%. In our cohort, lung cancer was the most frequent tumor.

### 2.1. Histological Diagnosis and In-House Molecular-Marker Evaluations

The histological diagnosis was performed at the Pathology Unit of the Campus Bio-Medico University Hospital Foundation (Rome, Italy), according to WHO criteria. Clinically relevant molecular predictive markers *(EGFR, KRAS, NRAS, BRAF, BRCA1, BRCA2, HRAS*, MSI, MMR, *c-KIT, PDGFRA, ALK, ROS1, HER2*, and PD-L1) were evaluated in-house in 111/184 patient samples, according to a clinical request to establish target therapies (i.e., non-small cell lung cancer (NSCLC), colon cancer, breast cancer, melanoma, etc.) For 73/184 patients, in-house molecular evaluations were not performed, because they were not requested for therapeutic purposes at the time of diagnosis (i.e., cholangiocarcinoma, testicular cancer, hepatocellular carcinoma, etc.).

In-house molecular-marker evaluations included the following predictive markers: *EGFR, KRAS, NRAS, BRAF, BRCA1, BRCA2, HRAS, c-KIT, PDGFRA, HER2* (single nucleotide variants (SNVs)), and *ALK* (SNVs). These were assayed with q-PCR or DNA-based NGS. MSI was assayed with q-PCR. *ALK* (rearrangements), *ROS−1, MET*, and *RET* were assayed with RNA-based NGS or RT-PCR. *ALK* (rearrangements), *ROS-1,* and *HER2* (amplification) were assayed with FISH. *ALK, ROS1,* MMR, and PD-L1 were assayed with immunohistochemistry. Different techniques were adopted based on the features of the tumor sample. Histological diagnosis and in-house molecular data were extracted from the clinical reports.

This study was conducted in accordance with the Declaration of Helsinki. The committee of Medical Ethics of the Campus Bio-Medico University Hospital Foundation (Roma, Italy) approved this study. The protocol code is 2015–003385-84.

### 2.2. Statistical Analysis

The difference in CGP Success Rates between primary neoplasms and metastatic samples was analyzed according to the Mann–Whitney U test. The statistical correlation between the CGP Success Rate and the SA was tested with a 2-sided Spearman test. The Kruskal–Wallis test was employed to evaluate the differences among sample types with different CGP Success Rates. All tests were two-sided, and a *p*-value <0.05 was considered statistically significant. All statistical analyses were performed with SPSS software (IBM SPSS Statistics 27, USA).

## 3. Results

### 3.1. Sample Data

In the present study, we analyzed 184 CGP reports of patients with solid tumors. The clinical–pathological variables we studied are described in Table 1. 

Nineteen different tumor types were analyzed with CGP. Among 184 samples, 81 (44%) were classified as primary cancers and 103 (56%) were classified as metastatic lesions. Among the metastatic lesions, 36/103 (35%) were lymph node metastases, and 65/103 (65%) were distant metastases. Lung cancer represented the most frequent entity (89/184; 48.4%), followed by breast cancer (36/184; 19.6%; Table 2).

Among the 184 samples, 109 (59.2%) were biopsies, 61 (33.2%) were surgical specimens, and 14 (7.6%) were cell-block specimens. According to the sample age categorization, 122/184 specimens were evaluated with CGP within <6 months, 21/184 were evaluated within 6–12 months, 25/184 were evaluated within 12–24 months, and 16/184 were evaluated >24 months. However, 34/184 CGP reports resulted in inconclusive results; therefore, the H&E sections of “inconclusive” cases were evaluated to determine the SA and TN%. A minority were classified as optimal samples (7/34), but most were either acceptable (13/34) or suboptimal samples (14/34).

### 3.2. CGP Data

The CGP analysis provided 150/184 satisfying reports (138/184 were completely satisfying and 12/184 were partially satisfying) and 34/184 inconclusive reports. The *Success Rate* was 81.5% (Table 3). No significant difference was found in *Success Rates* between primary and metastatic lesions (66/83 vs. 84/101; *p* = 0.553). 

In terms of “sample age”, ages ≤ 6 months had a *Success Rate* of 89.3%. The *Success Rates* were 76.2% for sample ages of 7–12 months, 64.0% for sample ages of 13–24 months, and 56.2% for sample ages >24 months. A significant negative correlation was found between the sample age and the *Success Rate* (*p* < 0.0001; r = − 0.302).

The CGP *Success Rate* was also evaluated in terms of sample type. *Success Rates* were 96.7% in surgical specimens, 74.3% in biopsies, and 71.4% in cell blocks. Thus, the *Success Rate* was significantly different among sample types (*p* = 0.001; Figure 1).

### 3.3. In-House Testing Compared to CGP

The routine in-house testing showed successful molecular analyses in 109/110 (99.1%) cases. The one failed analysis was also inadequate for the CGP analysis (EndoBronchial UltraSound-guided TransBronchial Needle Aspiration (EBUS-TBNA) NSCLC sample; SA: 9 mm^2^ and TN%: 20). The molecular test results showed that, in 85/109 cases, all tested genes were wild type, and in 24/109 samples, a targetable alteration was detected. These alterations were detected in 9 NSCLC samples (2 *ALK* rearrangements and 5 *EGFR* mutations, 1 *BRAF* mutation, and 1 G12C/*KRAS* mutation); 7 breast cancer samples (7 *HER2* amplifications); 5 colon cancer samples (5 *KRAS* mutations); and 3 gastric cancer samples (2 MSI-high and 1 *HER2* amplification). The CGP analyses confirmed the alterations detected in-house in 22/24 cases. Among the 2/24 alterations that CGP failed to detect, one was an *ALK* rearrangement in NSCLC, detected with IHC (ALK-D5F3) and with FISH (Vysis *ALK* Break Apart). For CGP, the specimen was suboptimal (EBUS-TBNA cell-block; SA: 9 mm^2^ and TN%: 30%). The other alteration that CGP failed to detect was a *KRAS* A146V mutation in metastatic colon cancer, detected with q-PCR. In this case, the specimen was acceptable for CGP (hepatic biopsy; SA: 25 mm^2^ and TN%: 50%).

CGP provided additional data on the TMB in 138/184 (74%) samples (all completely satisfying reports). TMB values varied between 0.00 mutations/Mb (mut/Mb) and 102.13 mut/Mb; the higher value was registered in a pleomorphic dermal sarcoma. In 30/138 (21.7%) samples, the TMB values were ≥10 mut/Mb; thus, in this group of patients, immunotherapy could be considered [7,8,9,10].

#### 3.3.1. Non-Small Cell Lung Cancer

Lung cancer was the most frequently observed tumor in our study; therefore, we analyzed the NSCLC cohort of patients further. We tested 88 NSCLC tissue samples with CGP. Among these, 50/88 (56.8%) were primary lesions and 38/88 (43.2%) were metastases. Completely or partially satisfying reports were obtained in 67/88 samples (76.3%) tested with CGP. Moreover, the CGP analyses revealed that 7/88 (8%) had *KRAS*/G12C mutations, 6/88 (6.8%) had *EGFR* mutations (1 ex.20-ins, 1 L858R, 1 G719S, 1 L861Q/G719A, 1 S768I/V769L, and 1 ex.19-del/T790M), 3/88 (3.4%) had *RET* fusions (1 RET RET-KIF5B; 1 RET RET-NCOA4; and 1 RET RET-CCDC6), 1/88 (1.1%) had a *BRAF* (V600E) mutation, and 1/88 (1.1%) had an *ALK* rearrangement. No *ROS1* rearrangements were found. Moreover, CGP detected 18 non-targetable mutations, including 13 in the *KRAS* gene (6 G12V; 2 G12F; 1 Q61L; 1 G12V; 1 G13V; 1 G13F; and 1 G12D and C185S), 3 in the *BRAF* gene (1 D594Y; 1 N581Y; and 1 G265R), and 2 in the *EGFR* gene (1 P753L and 1 H773_V774insNPH). CGP also provided TMB data in 59/88 (67%) samples (range: 0.00–42.87 mut/Mb). In 19/59 (32.2%) samples, the TMB was ≥10 mut/Mb.

Among 88 NSCLC samples, 72 had been tested in house. We found that 40/72 (55.6%) were primary tumors and 32/72 (44.4%) were metastases. Useful in-house molecular data were obtained for 71/72 samples (98.6%). 

All CGP reports matched the in-house molecular tests in terms of SNVs. Of note, CGP identified 1/2 (50%) of all *ALK* rearrangements that were detected with the in-house molecular approach. In this case, the CGP report declared a low exon coverage (hepatic biopsy; SA: 25 mm^2^ and TN%: 50). At the time of enrollment, therapy for *KRAS*/G12C, *RET*, and *NTRK* rearrangements was not the standard of care for patients with NSCLC. Consequently, the *KRAS*/G12C mutation, *RET* mutations, and *NTRK 1*/2/3 rearrangements were not evaluated with in-house testing.

#### 3.3.2. Breast Cancer

Breast cancer represented the second-most frequently observed tumor in our cohort. Of the 36 breast cancer samples tested with CGP, 11/36 (30.6%) were primary tumors and 25/36 (69.4%) were metastases. Completely or partially satisfying reports were obtained for 29/36 (80.6%) specimens. CGP analyses revealed 5/36 (13.9%) *HER2* amplifications and 7/36 (19.4%) targetable *PI3KCA* (5 H1047R; 2 E542K) mutations. The TMB was estimated in 27/36 (75%) samples (range: 0.00–11.35). In 2/27 (7.4%) samples, the TMB was ≥10 mut/Mb.

All breast cancer samples were tested in-house for *HER2* status with IHC or IHC/FISH. Of the 36 samples, 7 (19.4%) were classified as *HER2* positive. Of note, the CGP analysis failed to detect 2/7 (28.6%) *HER2* amplifications. Both samples were metastatic lesions (hepatic biopsy, SA: 5 mm^2^ and TN%: 5; bone biopsy, SA: 30 mm^2^ and TN%: 50).

At the time of enrollment, PIK3CA inhibitors were not the standard of care for patients with breast cancer. Consequently, *PIK3CA* mutations were not evaluated with in-house testing.

#### 3.3.3. Gastric Cancer

Of the four gastric cancers tested with CGP, all were derived from primary tumors. CGP detected MSI-high in 2/4 (50%) samples and provided information about the TMB (range: 0.00–80.69). In both these MSI-high gastric tumors, the TMBs were ≥10 mut/Mb (80.69 and 35.30 mut/Mb). Of the four gastric cancer samples, three were tested in-house for MSI and *HER2* status. In 2/3 (66.7%) samples, MSI-high was detected, and 1/3 samples (33.3%) was classified as *HER2* positive.

The CGP reports matched the in-house molecular results in terms of Microsatellite Stability/Microsatellite Instability (MSS/MSI). In 1/1 (100%) case, the CGP analysis failed to detect a *HER2* amplification (surgical specimen; SA: 200 mm^2^ and TN%: 40).

## 4. Discussion

In clinical studies, and for specific therapeutic subsets, CGP was proposed to offer personalized therapies in advanced or metastatic settings. In fact, it was suggested that CGP could reveal clinically relevant genetic alterations missed by conventional methods in up to 84% of cases [11,12]. Here, we investigated the feasibility and utility of the CGP approach in real-life samples from a prospective cohort of patients analyzed with F1CDx assays. Furthermore, we compared CGP results with data obtained from in-house molecular tests. In our series, 184 solid tumors were tested with the CGP approach. We found a *Success Rate* of 81.6%. Thus, in 23/184 samples, the CGP analysis failed to obtain the data revealed with molecular tests.

In the literature, the CGP *Success Rate* has varied greatly, from 68.8% to over 95% [13,14,15,16]. De Falco et al. investigated the feasibility of CGP in 122 patients and demonstrated a *Success Rate* of 68.8%. They showed that CGP results were influenced by the type of specimen (block vs. slides) and by the origin of the sample (surgery vs. biopsy) [14]. In contrast, Takeda et al. reported a very high CGP *Success Rate* (96.7%). In that study, the *Success Rate* was calculated after excluding samples classified as inadequate, based on a pathological evaluation at the Foundation Medicine centralized laboratory, and samples with an insufficient amount of DNA for genomic testing [13].

In our cohort, the *Success Rate* was highest in surgical specimens (96.7%) and lowest in biopsies (74.3%) and cell blocks (71.4%). Moreover, after grouping sample ages into four classes, we found that specimens with a sample age under 6 months had the greatest *Success Rate* (89.3%). Moreover, the *Success Rate* declined as the sample age increased (sample age 7–12 months: 76.2%; 13–24 months: 64.0%; >24 months: 56.2%). Similar data were reported by De Falco et al. in comparing samples grouped into two different sample ages (samples fixed within or before 5 years prior to the analysis) [14].

In the present study, the NSCLC subgroup had a *Success Rate* of 77%. However, the samples from advanced-stage NSCLC were typically obtained with minimally invasive procedures; thus, only a minority of the CGP analyses were based on an optimal surgical specimen; in fact, among the 88 NSCLC samples analyzed, 50 (56.8%) were biopsies, 25 (28.4%) were surgical specimens, and 13 (14.8%) were cell blocks. Nevertheless, all but one of the genomic results from in-house testing were confirmed with the CGP analysis. In particular, CGP failed to detect 1 of 2 *ALK* rearrangements diagnosed in-house. That finding was relevant, considering the low incidence of *ALK* rearrangements and their clinical relevance in NSCLC [17,18,19].

In breast and gastric cancer samples, CGP failed to detect *HER2* amplifications in 3/8 (37.5%) cases diagnosed with in-house testing (2/7 in breast cancer samples and 1/1 in a gastric cancer sample). However, some previous studies reported a high concordance between IHC/FISH and NGS/CGP. Thus, the lower agreement that we found could be related to heterogeneity in *HER2* expression. In fact, in prior studies, both intratumoral heterogeneity in *HER2* amplification and low tumor content in test specimens were found to limit NGS detection capabilities [20,21,22].

Currently, an IHC/FISH evaluation remains the first choice for assessing *HER2* status, according to ASCO/CAP 2018 guidelines. Indeed, *HER2*-targeted therapy was based on the predictive value of IHC/FISH for trastuzumab efficacy in breast cancer and gastroesophageal adenocarcinoma. In addition, NGS was proposed for gastric cancer when limited diagnostic tissue was available [23,24]. More studies are needed to investigate the concordance between NGS/CGP analyses and IHC/FISH evaluations in real life and to assess the best treatment for discordant cases.

The main limitation of the present study was the lack of an outcome analysis, due to the short observation time. Moreover, we analyzed a heterogeneous population of tumors that, if on one side represents the current clinical practice, on the other hand may impact on our results; for this reason, we performed a sub-population analysis to study the CGP performances in NSCLC and breast cancer independently. In addition, we lacked a comparison between in-house testing and CGP for *KRAS*/G12C, *RET* mutations, and *NTRK* rearrangements in NSCLC; and for *PIK3CA* mutations in breast cancer. These omissions were due to the lack of in-house evaluations, because at the time of enrollment, therapies for these markers were not the standard of care.

In conclusion, our findings, obtained from a real-life cohort of patients enrolled in a prospective clinical trial, suggested that tissue samples from surgical specimens that had been stored for less than six months should be preferred for CGP evaluations. The success of the CGP analysis was highest when the specimen requirements were respected and when the sample was optimal in terms of SA and the TN%. Therefore, based on our findings, we recommend that an in-house first-line molecular evaluation should be performed for all patients with solid tumors, and CGP should be performed as a complementary analysis in specific subsets of patients that are ineligible for available approved treatments and patients with an unexpected resistance to therapy.

## Figures and Tables

**Figure 1 diagnostics-13-00782-f001:**
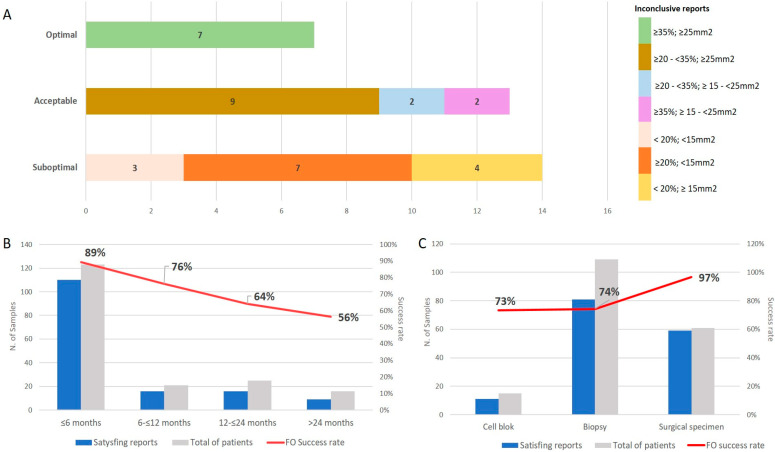
CGP *Success Rate* variation among sample types. (**A**) Inconclusive reports were evaluated as suboptimal, acceptable, or optimal samples. Colors represent the different combinations of tumor surface area (mm^2^) and the percentage of tumor nuclei (%). (**B**) Satisfying reports (completely and partially satisfying) are related to sample age (months). Red line represents the *Success Rate*. (**C**) Satisfying reports (completely and partially satisfying) are related to sample type (cell-block, biopsy, or surgical specimen). Red line represents the *Success Rate*.

**Table 1 diagnostics-13-00782-t001:** Baseline characteristics of patients with solid tumors.

Variable	N
Patients	184
Male	84
Female	100
Male to female ratio	1:1.2
Age range, y	24–84
Median age at diagnosis, y	75

**Table 2 diagnostics-13-00782-t002:** Tumor types analyzed with comprehensive genomic profiling.

Origin	Primitive	Metastasis	Total (%)
Lung:	50	39	89 (48.4)
70 Adenocarcinoma			
11 Squamous cell carcinoma			
1 Adenosquamous			
1 SCLC			
1 LCNC			
5 NOS			
Breast	10	26	36 (19.6)
Colon	1	11	12 (6.5)
Soft tissues (sarcoma)	10	0	10 (5.4)
Pancreas	3	2	5 (2.7)
Stomach	4	0	4 (2.2)
Thyroid	1	3	4 (2.2)
Skin (Melanoma)	0	4	4 (2.2)
Testis (Germ Cell Tumor)	0	3	3 (1.6)
Liver (HCC)	1	1	2 (1.1)
Bile ducts (Cholangiocarcinoma)	2	0	2 (1.1)
Salivary Gland	1	1	2 (1.1)
Thymus	1	0	1 (0.5)
Nasopharynx	0	1	1 (0.5)
Kidney	0	1	1 (0.5)
Esophagus	1	0	1 (0.5)
Mesothelium	0	1	1 (0.5)
Penis	1	0	1 (0.5)
Unknown origin	-	5	5 (2.7)
Total	86	98	184 (100)

SCLC: small-cell lung cancer; LCNC: large-cell neuroendocrine carcinoma; NOS: not otherwise specified (comprising the diagnosis of NSCLC, NOS, and large-cell lung cancer); HCC: hepatocellular carcinoma.

**Table 3 diagnostics-13-00782-t003:** Satisfying and inconclusive comprehensive genomic profile reports.

CGP reports	Category	N	Total	
Satisfying reports	Completely satisfying	138	150	
	Partially satisfying	12		
Inconclusive reports			34	
*Inconclusive reports*				
Specimen class	SA (mm^2^)	TN% (%)	N	Total
Suboptimal	<15	<20	3	14
	<15	≥20	7	
	≥15	<20	4	
Acceptable	≥25	≥20–<35	9	13
	≥15–<25	≥20–<35	2	
	≥15–<25	≥35	2	
Optimal	≥25	≥35	7	7
*Success Rates*				
Variable	Satisfying reports (N)	Total patients (N)	Success Rate (%)	
Sample age (months)				
≤6	109	122	89.3	
6–≤12	16	21	76.2	
12–≤24	16	25	64.0	
>24	9	16	56.2	
Type of specimen				
Cell blocks	10	14	71.4	
Biopsies	81	109	74.3	
Surgical specimens	59	61	96.7	

## Data Availability

Not applicable.

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
