# Peer review of "Feasibility of Comprehensive Genomic Profiling (CGP) in Real-Life Clinical Practice"

_diagnostics, 2023, doi:10.3390/diagnostics13040782_

Round 1

Reviewer 1 Report

The overall study is well presented and the novelty of the paper is high. As a personal recommendation, despite the fact that liquid biopsy is now one of the hot topics in cancer research, cfDNA or ctDNA samples can also be analyzed in cancer patients 

Author Response

The reviewer wrote: As a personal recommendation, despite the fact that liquid biopsy is now one of the hot topics in cancer research, cfDNA or ctDNA samples can also be analyzed in cancer patients.

Author answer: We thanks for suggestion. Actually, the cfDNA or ctDNA analysis was not performed in the prospective clinical trial GO40782 - RXDX-101-02 and for this reason we do not have data about. Nevertheless, we will keep this point in consideration for further research.

Reviewer 2 Report

The title reflects the subject of the manuscript. It presents a clear and useful message for clinical practice. It is well written in terms of clarity, style, and use of English language. The discussion section is sufficiently detailed and explains adequately the purpose of this study in the context of published information. The conclusion accurately and clearly explains the main result. The length of the manuscript is good. All references are appropriate and current.

Minor points:

- I would advise trasnfering "In the present study, we analyzed 184 GCP reports of patients with solid tumors. The 75 clinical-pathological variables we studied are described in Table 1." as well as table 1 in the results section.

- In the methods section better refer as "Statistical analysis" rather than statistics

- Figure 1a could potentially be omitted as it illustrates really little information.

- The limitations section needs to be improved. The heterogeneity of types of tumors and subsequently variable number of cases per cancer may highly impact outcomes etc.

Author Response

1) The reviewer wrote: I would advise trasnfering "In the present study, we analyzed 184 GCP reports of patients with solid tumors. The 75 clinical-pathological variables we studied are described in Table 1." as well as table 1 in the results section.

Author answer: as the reviewer suggested, we transferred the sentence and the table 1 in results section.

2) The reviewer wrote: In the methods section better refer as "Statistical analysis" rather than statistics

Author answer: as the reviewer suggested, we changed "statistics" in "Statistical analysis".

3) The reviewer wrote: Figure 1a could potentially be omitted as it illustrates really little information.

Author answer: as the reviewer suggested, we modified Figure 1 omitting the pie chart.

4) The reviewer wrote: The limitations section needs to be improved. The heterogeneity of types of tumors and subsequently variable number of cases per cancer may highly impact outcomes etc.

Author answer: as the reviewer suggested, we added the following sentence in limitation section "Moreover, we analyzed a heterogeneous population of tumors that, if on one side represents the current clinical practice, on the other hand may impact on our results; for this reason, we performed a sub-population analysis to study the CGP performances in NSCLC and breast cancer independently"

Reviewer 3 Report

The authors Nibid et al. conducted a study on comprehensive gene profiling in different solid tumors in their institution. They analyzed the quality of specimens and compared the results of comprehensive gene profiling to other types of molecular testing such as  FISH or IHC. They found comparable results to other studies and mainly found that small non-invasive specimens and older specimens result in lower success of the testing methods. 

The topic of molecular tumor boards and standardized panel testing is rising within recent years. Therefore, studies performed on methodology are still important. The main limitations of the study are already discussed in the manuscript. It will be needed to collect outcome data in the future to assess the methods of systematic molecular testing in solid tumors. 

Author Response

The reviewer wrote: The main limitations of the study are already discussed in the manuscript. It will be needed to collect outcome data in the future to assess the methods of systematic molecular testing in solid tumors. 

Author answer: we thanks for the suggestion. The aim of the paper was to evaluate the (laboratory) feasibility of CGP as a standard of care for solid tumours. However, we will keep this valuable suggestion in mind for the planning of future studies.